# Global burden of subarachnoid hemorrhage among adolescents and young adults aged 15–39 years: A trend analysis study from 1990 to 2021

Xuanchen Liu[1,2,3◦], Rui Cheng[1,2,3◦], Yingda Song[4◦], Xiaoxiong Yang[1,2], Xiaochen Niu[1,2,3], Chunhong Wang[1,2], Guijun Jia[1,2]*, Hongming Ji[1,2,3]*

1 The Neurosurgery Department of Shanxi Provincial People's Hospital, Shanxi Medical University, Taiyuan, Shanxi Province, China, 2 The Neurosurgery Department of Shanxi Provincial People's Hospital, Taiyuan, Shanxi Province, China, 3 Shanxi Provincial Key Laboratory of Intelligent, Big Data and Digital Neurosurgery, Taiyuan, Shanxi Province, China, 4 Shanxi Provincial People's Hospital, Taiyuan, Shanxi Province, China

◦ These authors contributed equally to this work.
* hongmingj@sina.com (HJ); niuniujia2@163.com (GJ)

**Data Availability Statement:** All relevant data are included within the manuscript and its Supporting information files.

## Abstract

### Objective

This study aims to analyze the global burden of subarachnoid hemorrhage (SAH) among adolescents and young adults (AYAs) aged 15–39 years from 1990 to 2021, highlighting spatial and temporal trends and providing insights for future public health strategies.

### Methods

Data were collected from the Global Burden of Disease Study 2021 (GBD 2021), which includes comprehensive evaluations of health conditions and associated risk factors across 204 countries and territories. The focus was on SAH incidence, prevalence, mortality, and disability-adjusted life years (DALYs) among AYAs. The data were segmented by age groups (15–19, 20–24, 25–29, 30–34, 35–39 years) and socio-demographic index (SDI) quintiles. Statistical analyses, including Joinpoint regression and decomposition analysis, were employed to assess temporal trends and the impact of population growth, aging, and epidemiological changes.

### Results

From 1990 to 2021, the global number of SAH incident cases among AYAs increased by 12.6%, from 109,120 cases in 1990 to 122,822 cases in 2021. Prevalent cases rose by 17.1%, from 1,212,170 cases in 1990 to 1,419,127 cases in 2021. Conversely, the number of deaths decreased by approximately 26.6%, from 30,348 cases in 1990 to 22,266 cases in 2021. Similarly, DALYs decreased by 23.7%, from 1,996,041 cases in 1990 to 1,523,328 cases in 2021. Notably, over these thirty years, the age-standardized rates (ASR) of incidence, prevalence, mortality, and DALYs for the AYA population showed an overall decreasing trend, despite fluctuations in specific periods. The age-standardized mortality

**Funding:** This work was funded by the Basic Research Plan of Shanxi Province in 2021 (202103021224386) and the 2024 Graduate Education Innovation Plan of Shanxi Province (2024SJ177). The funders had no role in study design, data collection and analysis, decision to publish, or preparation of the manuscript. There was no additional external funding received for this study.

**Competing interests:** The authors have declared that no competing interests exist.

rate (ASMR) and age-standardized DALYs (ASR for DALYs) decreased continuously with an average annual percentage change (AAPC) of -2.2% (95% CI: -2.36, -2.04) and -2.02% (95% CI: -2.17, -1.88), respectively. The age-standardized incidence rate (ASIR) and age-standardized prevalence rate (ASPR) had an AAPC of -0.8% (95% CI: -0.85, -0.75) and -0.65% (95% CI: -0.66, -0.64), respectively. Particularly, the ASIR showed a continuous decline from 1990 to 2015, followed by a slight increase from 2014 to 2019 (APC: 0.14%, 95% CI: 0.03, 0.25), and accelerated growth from 2019 to 2021 (APC: 1.23%, 95% CI: 0.88, 1.57). The ASPR declined from 1990 to 2019, followed by an increase from 2019 to 2021 (APC: 0.15%, 95% CI: 0.05, 0.25). Regional analysis revealed substantial burdens in the Middle-SDI and Low-Middle-SDI regions, with the Middle-SDI region having the highest incidence, prevalence, mortality, and DALYs. Decomposition analysis indicated that population growth was the primary driver of increased SAH cases, while epidemiological changes contributed significantly to the decline in deaths and DALYs.

## Conclusion

The findings underscore the need for targeted public health interventions, particularly in low and low-middle-SDI regions, to reduce the burden of SAH among AYAs. Improved healthcare resources, enhanced health education, and preventive strategies are crucial. This study provides valuable data to inform future public health policies and resource allocation, emphasizing the importance of addressing the unique challenges faced by AYAs.

## Introduction

Subarachnoid hemorrhage (SAH) is a type of stroke characterized by bleeding into the subarachnoid space, the area between the brain and its covering tissues1 [1]. This condition predominantly affects adults and is a significant cause of morbidity and mortality. SAH is often classified as a non-traumatic stroke subtype, distinct from other forms of hemorrhagic stroke due to its unique pathophysiology and clinical presentation. While SAH primarily affects individuals in their middle to late adult years, it can also occur in younger populations [2]. Adolescents and young adults (AYAs), defined as individuals aged 15 to 39 years, are particularly vulnerable due to various risk factors, including genetic predispositions, lifestyle choices, and underlying health conditions [3]. The incidence and impact of SAH in this age group are of increasing concern, given the complex economic and psychological pressures they face [4].

SAH is one of the major acute cerebrovascular diseases in neurosurgery, with high mortality and disability rates, leading to a substantial health burden. The World Health Organization (WHO) includes SAH under the broader category of stroke in its Global Burden of Disease (GBD) assessments [5]. Despite its significant impact, there has been a lack of systematic studies focusing on the global burden of SAH, with most research treating it as a subset of stroke [6]. This approach has resulted in a gap in understanding the distinct epidemiological and clinical characteristics of SAH, particularly among the AYAs population. AYAs are increasingly facing complex economic and psychological pressures, exacerbating their vulnerability to cerebrovascular diseases and imposing a significant socio-economic burden globally [7]. This age group faces unique challenges, including the onset of independent living, educational pursuits, and career establishment, which can influence health behaviors and access to care [8]. Understanding the burden of SAH in this age group is critical for designing effective prevention and treatment programs that address the specific needs of AYAs.

Our study utilizes the 2021 Global Burden of Disease (GBD) database, providing the most comprehensive and current estimates of subarachnoid hemorrhage (SAH) burden across all 204 countries and territories [9]. Unlike previous studies limited to pre-2019 data, this research offers post-COVID-19 insights and focuses specifically on the adolescent and young adult (AYA) population.

This study systematically describes the burden of SAH among AYAs in 2021, including incidence, prevalence, mortality, and disability-adjusted life years (DALYs), with a focus on regional and temporal trends. By utilizing the GBD statistical model, we aim to provide a detailed analysis that highlights the significant public health challenge posed by SAH and offers insights for improving healthcare strategies and resource allocation.

## Materials and methods

### Data acquisition and download

The GBD 2021 represents a seminal update in a lineage of annual studies initiated in 1990, dedicated to quantifying the worldwide impact of diseases, injuries, and associated risk factors. Offering standardized evaluations for over 369 health conditions and 88 risk elements across 204 nations and territories, the GBD 2021 presents a panoramic view of global health challenges. In this framework, SAH, characterized by bleeding within the brain's subarachnoid space, is classified as a non-traumatic stroke subtype. Diagnosis and categorization of SAH incidents rely on International Classification of Diseases (ICD) codes I60 (ICD-10), aligning SAH within the highly specific Level 4 category, nested under non-communicable diseases, cardiovascular diseases, and stroke classifications [10]. AYAs, as defined by the United Nations and widely adopted in health research, encompass individuals aged 15 to 39.

The GBD data pertinent to this age group is conveniently accessible through the Global Health Data Exchange (GHDX) query platform, enabling researchers to extract and scrutinize detailed metrics, including incidence, prevalence, mortality, and DALYs for SAH and other pertinent health issues affecting AYAs. Our investigation focused on acquiring comprehensive measurements of SAH's health consequences among AYAs, complete with 95% uncertainty intervals (UI) [11]. This encompassed incidence, mortality, prevalence, and DALYs. Leveraging the 2021 GBD dataset, our analysis spanned the entire globe, segmented by five sociodemographic development tiers, 21 predefined GBD regions, and encompassing every one of the 204 countries and territories, with no gender bias. The analysis covered the period from 1990 to 2021, meticulously dividing the data into 5-year age brackets to capture the unique profiles of AYAs aged 15–19, 20–24, 25–29, 30–34, and 35–39.

In GBD 2021, the Sociodemographic Index (SDI) was computed to embody the composite level of health-correlated social and economic contexts across regions. This SDI is derived as the geometric mean of indices ranging from 0 to 1, encompassing the total fertility rate among females under the age of 25, the mean years of education completed by individuals aged 15 and above, and the country's lag-distributed income per capita. The 204 countries included in GBD 2021 were then stratified into quintiles—labeled as low, low-middle, middle, high-middle, and high—based on their respective national SDI estimates for the year 2021 [11].

### Statistical analysis

Data on the incidence, prevalence, mortality, and DALYs of SAH were extracted from the GBD 2021. In this study, ages in the range of 15–39 years were divided into five GBD age groups at 5-year intervals. All data calculations were performed using age-standardization. Age-standardized data for incidence, mortality, prevalence, and DALYs were downloaded from the GBD database. The purpose of age-standardization is to remove the effect of

differences in age structure. This study adopts the Joinpoint regression analysis model to assess the temporal trends in disease incidence or mortality rates. Joinpoint regression is a statistical method used to analyze trends in time-series data, such as the incidence, prevalence, mortality, and DALYs. This method partitions the continuous time-series data's linear regression model into several statistically significant trend segments, with each portion described by a linear model to identify the optimal number of turning points. The method uses a Monte Carlo permutation test to determine the optimal number of joinpoints and their locations based on the statistical significance of the change in trend [12]. For each segment, the Joinpoint regression model calculates the annual percentage change (APC) in the outcome variable and the 95% confidence intervals for the APC. The average annual percentage change (AAPC) is calculated over the entire study period.

This approach serves to define the incidence trends within specific timeframes by meticulously examining the changing patterns of these health metrics.

The AAPC values for deaths and DALYs were estimated using a linear regression model. Hierarchical cluster analysis was then performed based on the AAPC values to evaluate the patterns of disease burden changes across 21 regions and 204 countries and territories. At the regional level, all regions were classified into four categories: remained stable or minor decrease, moderate decrease, substantial decrease, and significant decrease. At the country level, the 204 countries and territories were similarly categorized into four groups: significant increase, minor increase, remained stable or minor decrease, and significant decrease.

To quantify the drivers of changes in the number of individuals with SAH, we employed decomposition analysis to estimate the relative contributions of population growth, aging, and epidemiological change. This method, developed by Das Gupta and widely applicable in epidemiological research, offers a principled dissection of how each component uniquely contributes to the overall trend [13]. It summarizes the contribution of various factors to observed changes by algebraically isolating the standardized impact of each contributing multiplicative factor. Epidemiological change involves changes in the incidence or prevalence of disease due to various factors, such as improvements in healthcare, the introduction or evolution of risk factors, the effectiveness of prevention strategies, advancements in treatment, or changes in environmental or lifestyle factors.

The Bayesian age-period-cohort (BAPC) model was employed to predict the age-standardized rates (ASR) and the number of SAH cases among AYAs from 2022 to 2040 [14]. By utilizing integrated nested Laplace approximations (INLA), the BAPC model approximated the marginal posterior distributions, effectively circumventing the mixing and convergence issues typical of traditional Bayesian methods that rely on Markov chain Monte Carlo (MCMC) techniques.

Data cleaning, computation, and graph plotting were conducted using R software (version 4.3.2) in this study, a two-sided $p$-value $<0.05$ was defined as the significance threshold.

## Ethics statement

This study utilized data exclusively obtained from the Global Burden of Disease (GBD) 2021 database, a publicly available resource managed by the Institute for Health Metrics and Evaluation (IHME). The data used in this research are anonymized and aggregated, and do not contain any personally identifiable information. As such, this study meets the criteria for exemption from full ethical review according to the guidelines provided by Shanxi Provincial People's Hospital Ethics Committee and the University of Washington's IRB, which has oversight over the GBD project.

## Results

### Global trends

From 1990 to 2021, the number of incidents and prevalent cases of SAH among AYA increased significantly worldwide, while the number of DALYs and deaths decreased markedly. Specifically, the number of prevalent cases rose from 1,212,170 in 1990 (95% UI: 1,023,467 to 1,444,829) to 1,419,126 in 2021 (95% UI: 1,219,651 to 1,652,774), a 17.1% increase. Incident cases grew from 109,120 in 1990 (95% UI: 68,971 to 156,908) to 122,821 in 2021 (95% UI: 79,455 to 172,469), representing a 12.6% rise. In contrast, the number of deaths decreased from 30,347 in 1990 (95% UI: 24,555 to 36,029) to 22,266 in 2021 (95% UI: 18,642 to 27,939), a 26.6% reduction. DALYs decreased from 1,996,041 in 1990 (95% UI: 1,655,848 to 2,347,834) to 1,523,328 in 2021 (95% UI: 1,291,523 to 1,859,045), a 23.7% decline (Table 1, S1 Table).

Over these thirty years, the ASR of incidence, prevalence, mortality, and DALYs in the AYA population generally showed a decreasing trend, despite fluctuations during specific periods. The age-standardized mortality rate (ASMR) and age-standardized DALYs rate (ASDR) continuously declined, with an average annual percentage change (AAPC) of -2.2 (95% CI: -2.36, -2.04) and -2.02 (95% CI: -2.17, -1.88), respectively (Table 1). The age-standardized incidence rate (ASIR) exhibited an AAPC of -0.80% (95% CI: -0.85%, -0.75%), while the age-standardized prevalence rate (ASPR) demonstrated an AAPC of -0.65% (95% CI: -0.66%, -0.64%) over the study period. Notably, the ASIR showed a continuous decline from 1990 to 2015, followed by a slight increase from 2014 to 2019 (APC: 0.14, 95% CI: 0.03, 0.25), and accelerated growth from 2019 to 2021 (APC: 1.23, 95% CI: 0.88, 1.57) (Fig 1A). The ASPR declined from 1990 to 2019, then increased from 2019 to 2021 (APC: 0.15, 95% CI: 0.05, 0.25) (Fig 1B).

By 2021, the ASIR was 4.06 per 100,000 population, a decrease of 0.25 cases compared to 1990. The ASPR was 46.95 per 100,000 population, also a decrease of approximately 0.25 cases compared to 1990. In the same year, the ASMR decreased to 0.73 per 100,000 population, a reduction of 0.48 cases compared to 1990. The global age-standardized DALYs for the AYA population was 50.3 per 100,000 population in 2021, a decrease of 0.49 cases compared to 1990 (Table 1).

When analyzing sex-specific data, men among AYAs experienced a heavier burden of SAH. Initially, the incidence cases for men and women were comparable, but by 2021, the rate for men was 1.14 times higher. Similarly, the ASIR for men and women was initially similar, but by 2021, it was 1.11 times higher for men. Interestingly, the global prevalence of SAH shifted from 1990 to 2021. In 1990, the prevalence was slightly higher in women (Male: 587,556 [95% UI: 495,233–700,174], Female: 624,615 [95% UI: 527,893–743,075]). By 2021, this trend reversed, with the prevalence being slightly higher in men (Male: 711,938 [95% UI: 612,057–829,942], Female: 707,189 [95% UI: 608,193–822,446]) (Table 1).

Using Joinpoint regression analysis, several key years were identified as points of trend shifts, including ASIR in 1995, 1998, 2010, 2014, and 2019; ASPR in 1999, 2005, 2009, 2012, and 2019; ASMR in 1994, 2000, and 2005; and ASR for DALYs in 1994, 2000, 2005, and 2013. These time points mark significant changes in the trends of health burdens (Fig 1, S1 Table).

### Global trends by region

In 2021, our study across different global regions indicated that the burden of SAH among AYAs was most substantial in the Middle-SDI and Low-Middle-SDI regions. The Middle-SDI region reported the highest numbers of incidence, prevalence, mortality, and DALYs, with the respective figures being 40,732 (95% UI: 26,362 to 57,205), 474,515 (95% UI: 406,691 to 553,358), 7,624 (95% UI: 6,426 to 8,697), and 519,996 (95% UI: 445,139 to 585,764) (Table 1).

**Table 1. Global trends of SAH in AYAs from 1990 to 2021.**

**Global trends on incidence of SAH among AYAs from 1990 to 2021**

| | ASIR,1990 | ASIR,2021 | Percentage changes (%) | AAPCs | Incident cases,1990 | Incident cases,2021 | Percentage changes (%) |
|---|---|---|---|---|---|---|---|
| Global | 5.19(3.28–7.45) | 4.05(2.62–5.7) | -0.22 | -0.8(-0.85 to -0.75)* | 109120(68971–156907) | 122822(79455–172469) | 0.13 |
| **Sex** | | | | | | | |
| Male | 5.23(3.31–7.63) | 4.27(2.8–6.11) | -0.18 | -0.66(-0.7 to -0.62)* | 55580(35177–81251) | 65529(43033–93639) | 0.18 |
| Female | 5.15(3.21–7.44) | 3.83(2.41–5.45) | -0.26 | -0.97(-1.02 to -0.91)* | 53540(33351–77459) | 57293(36111–81430) | 0.07 |
| **SDI** | | | | | | | |
| High SDI | 4.63(2.94–6.61) | 3.56(2.23–5.1) | -0.23 | -0.84(-0.91 to -0.77)* | 16706(10662–23810) | 13625(8610–19387) | -0.18 |
| High-middle SDI | 5.27(3.36–7.55) | 3.86(2.49–5.5) | -0.27 | -1.01(-1.09 to -0.93)* | 23765(15139–34006) | 18607(12090–26272) | -0.22 |
| Middle SDI | 5.64(3.54–8.13) | 4.23(2.73–5.96) | -0.25 | -0.93(-0.99 to -0.87)* | 39713(24858–57400) | 40732(26362–57205) | 0.03 |
| Low-middle SDI | 5.28(3.36–7.49) | 4.52(2.98–6.29) | -0.14 | -0.52(-0.56 to -0.47)* | 21996(13996–31312) | 35243(23227–49188) | 0.60 |
| Low SDI | 4.08(2.5–5.97) | 3.54(2.25–5.08) | -0.13 | -0.44(-0.48 to -0.4)* | 6827(4183–10034) | 14502(9175–20914) | 1.12 |

**Global trends on prevalence of SAH among AYAs from 1990 to 2021**

| | ASIR,1990 | ASIR,2021 | Percentage changes (%) | AAPCs | Incident cases,1990 | Incident cases,2021 | Percentage changes (%) |
|---|---|---|---|---|---|---|---|
| Global | 57.4(48.56–68.28) | 46.95(40.31–54.73) | -0.18 | -0.65(-0.66 to -0.64)* | 1212170(1023467–1444829) | 1419127(1219651–1652774) | 0.17 |
| **Sex** | | | | | | | |
| Male | 54.98(46.44–65.38) | 46.51(39.95–54.27) | -0.15 | -0.76(-0.78 to -0.74)* | 587556(495233–700174) | 711938(612057–829943) | 0.21 |
| Female | 59.87(50.7–71.09) | 47.39(40.72–55.18) | -0.21 | -0.54(-0.55 to -0.53)* | 624615(527893–743075) | 707189(608193–822446) | 0.13 |
| **SDI** | | | | | | | |
| High SDI | 52.92(44.93–62.68) | 48.29(41.47–56.49) | -0.09 | -0.3(-0.33 to -0.28)* | 190009(161654–224574) | 182882(157891–212832) | -0.04 |
| High-middle SDI | 59.52(50.3–70.84) | 46.63(40.15–54.14) | -0.22 | -0.79(-0.82 to -0.76)* | 268182(226570–319296) | 223359(193262–257844) | -0.17 |
| Middle SDI | 63.2(53.31–75.47) | 49.39(42.26–57.7) | -0.22 | -0.8(-0.82 to -0.78)* | 448742(377474–537152) | 474515(406691–553358) | 0.06 |
| Low-middle SDI | 53.51(45.32–63.41) | 46.47(39.97–54.18) | -0.13 | -0.46(-0.47 to -0.44)* | 224959(189729–267715) | 363469(312164–424424) | 0.62 |
| Low SDI | 46.56(39.49–55.23) | 41.93(35.97–48.92) | -0.1 | -0.34(-0.35 to -0.33)* | 78967(66577–94183) | 173499(147938–203896) | 1.20 |

**Global trends on mortality of SAH among AYAs from 1990 to 2021**

| | ASIR,1990 | ASIR,2021 | Percentage changes (%) | AAPCs | Incident cases,1990 | Incident cases,2021 | Percentage changes (%) |
|---|---|---|---|---|---|---|---|
| Global | 1.45(1.17–1.72) | 0.73(0.61–0.92) | -0.5 | -2.2(-2.36 to -2.04)* | 30348(24555–36029) | 22266(18642–27939) | -0.27 |
| **Sex** | | | | | | | |
| Male | 1.64(1.08–2.13) | 0.88(0.66–1.25) | -0.46 | -2.02(-2.2 to -1.84)* | 17392(11459–22634) | 13568(10287–19336) | -0.22 |
| Female | 1.26(0.93–1.47) | 0.57(0.48–0.71) | -0.55 | -2.49(-2.63 to -2.36)* | 12956(9633–15192) | 8698(7343–10715) | -0.33 |
| **SDI** | | | | | | | |

*(Continued)*

**Table 1.** (Continued)

| | | | | | | | |
|---|---|---|---|---|---|---|---|
| High SDI | 1.19(1.13–1.25) | 0.47(0.45–0.5) | -0.61 | -2.89(-3.25 to -2.54)* | 4332(4132–4574) | 1872(1778–1966) | -0.57 |
| High-middle SDI | 1.43(1.17–1.66) | 0.55(0.49–0.63) | -0.62 | -3.06(-3.42 to -2.7)* | 6458(5284–7505) | 2774(2473–3169) | -0.57 |
| Middle SDI | 1.76(1.31–2.08) | 0.78(0.66–0.89) | -0.56 | -2.6(-2.74 to -2.46)* | 12300(9111–14531) | 7624(6426–8697) | -0.38 |
| Low-middle SDI | 1.41(0.96–1.98) | 0.94(0.69–1.28) | -0.33 | -1.29(-1.47 to -1.11)* | 5852(3993–8194) | 7316(5431–10007) | 0.25 |
| Low SDI | 0.82(0.42–1.5) | 0.65(0.33–1.34) | -0.21 | -0.75(-0.81 to -0.69)* | 1373(698–2511) | 2653(1360–5476) | 0.93 |

**Global trends on DALYs of SAH among AYAs from 1990 to 2021**

| | ASIR,1990 | ASIR,2021 | Percentage changes (%) | AAPCs | Incident cases,1990 | Incident cases,2021 | Percentage changes (%) |
|---|---|---|---|---|---|---|---|
| Global | 94.82(78.74–111.49) | 50.33(42.65–61.48) | -0.47 | -2.02(-2.17 to -1.88)* | 1996041(1655848–2347834) | 1523328(1291523–1859045) | -0.24 |
| **Sex** | | | | | | | |
| Male | 105.13(72.27–134.65) | 58.51(45.42–80.66) | -0.44 | -1.91(-2.07 to -1.75)* | 1121351(769088–1437249) | 896764(696593–1234646) | -0.20 |
| Female | 84.27(65.36–97.61) | 41.95(36.13–50.56) | -0.5 | -2.23(-2.34 to -2.11)* | 874690(676756–1013791) | 626564(540351–755220) | -0.28 |
| **SDI** | | | | | | | |
| High SDI | 77.53(73.07–82.1) | 35.03(32.51–37.82) | -0.55 | -2.51(-2.88 to -2.14)* | 280220(264207–296606) | 135255(125752–145826) | -0.52 |
| High-middle SDI | 95.64(79.5–109.79) | 40.71(36.33–46.12) | -0.57 | -2.74(-3.17 to -2.32)* | 430510(357831–494195) | 198948(177774–224887) | -0.54 |
| Middle SDI | 114.44(88.08–133.6) | 54.01(46.28–60.83) | -0.53 | -2.39(-2.51 to -2.28)* | 807707(620995–943361) | 519996(445139–585764) | -0.36 |
| Low-middle SDI | 91.13(64–124.25) | 61.99(47.33–82.3) | -0.32 | -1.2(-1.32 to -1.08)* | 382202(268594–520197) | 484031(369603–642989) | 0.27 |
| Low SDI | 55.17(30.78–95.36) | 44.43(25.24–85.63) | -0.19 | -0.71(-0.76 to -0.66)* | 93280(52053–162128) | 183366(104280–355005) | 0.97 |

**Notes:** This table presents a comprehensive analysis of the global trends in SAH among AYAs aged 15–39 years from 1990 to 2021. The data is segmented by ASIR, ASPR, ASMR, and ASDR, along with the corresponding incident, prevalence, mortality, and DALY cases. The table also provides percentage changes and AAPCs over the study period. The data is further broken down by global, sex, and SDI categories. Numbers marked with an asterisk * indicate p < 0.05, signifying statistical significance.

**Abbreviations:** ASIR: Age-Standardized Incidence Rate; ASPR: Age-Standardized Prevalence Rate; ASMR: Age-Standardized Mortality Rate; ASDR: Age-Standardized Disability-Adjusted Life Year Rate; AAPC: Average Annual Percentage Change; SDI: Sociodemographic Index; UI: Uncertainty Interval

Among the ASR, the Low-Middle-SDI region exhibited the highest ASIR, ASMR, and ASDR, with values of 4.52 (95% UI: 2.98 to 6.30), 0.94 (95% UI: 0.70 to 1.29), and 62.00 (95% UI: 47.34 to 82.31), respectively.

Over the 31 years, the ASIR, ASPR, ASMR, and ASDR across different SDI regions showed a declining trend. The High-Middle SDI region demonstrated the fastest declines in ASIR, ASMR, and ASDR, with average annual percentage changes (AAPC) of -1.01 (95% CI: -1.09 to -0.93), -3.06 (95% CI: -3.42 to -2.70), and -2.74 (95% CI: -3.17 to -2.32), respectively (Table 1).

At the regional level, from 1990 to 2021, East Asia experienced the largest declines in ASPR, ASMR, and ASDR, with average annual percentage changes of -1.26 (95% CI: -1.28 to -1.23), -4.22 (95% CI: -4.54 to -3.89), and -3.75 (95% CI: -4.01 to -3.49), respectively. Notably, South Asia exhibited the highest numbers in incidence, prevalence, deaths, and DALYs (S2 Table).

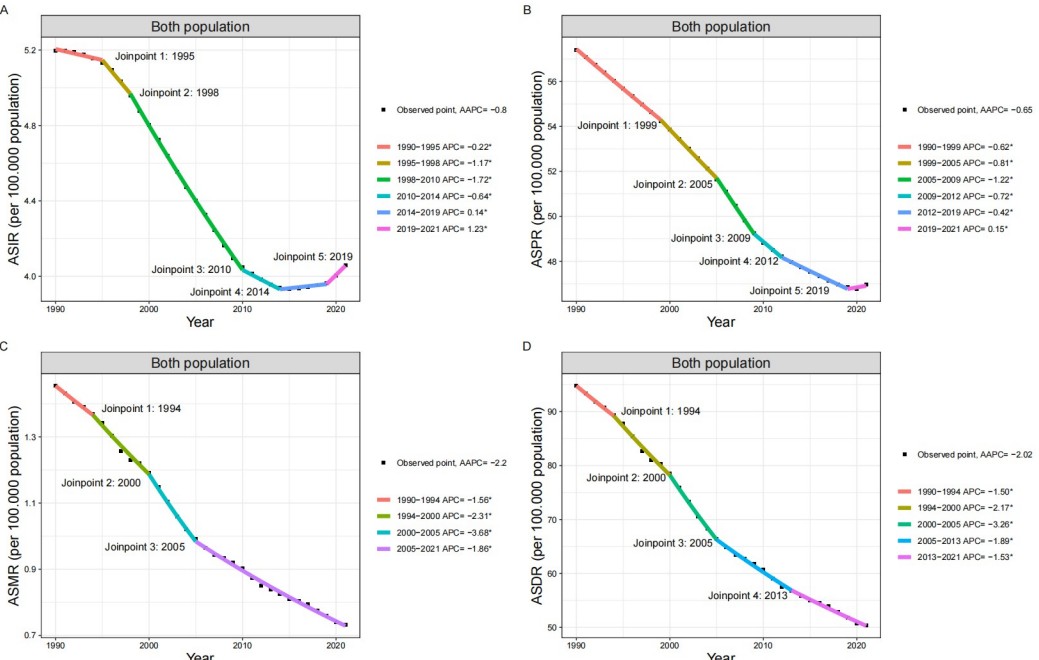

**Fig 1. Global trends in SAH among AYAs from 1990 to 2021 using joinpoint regression analysis. Notes:** This image displays the age-standardized rates of incidence, prevalence, death, and DALYs for SAH among AYAs globally from 1990 to 2021. **A:** Global Incidence of SAH in AYAs. **B:** Global Prevalence of SAH in AYAs. **C:** Global Mortality of SAH in AYAs. **D:** Global DALYs of SAH in AYAs. **Abbreviations:** ASIR: Age-Standardized Incidence Rate; ASPR: Age-Standardized Prevalence Rate; ASMR: Age-Standardized Mortality Rate; DALYs: disability-adjusted life years; ASDR: Age-Standardized Disability-Adjusted Life Year Rate; AAPC: Average Annual Percentage Change; SDI: Sociodemographic Index; UI: Uncertainty Interval.

## Cluster analysis

To identify regions with similar patterns of change in SAH burden among the AYA population, a hierarchical cluster analysis was performed across 21 regions. The regions were classified into four categories based on the AAPC in rates for SAH-related deaths and DALYs. Eastern Europe and Southern Sub-Saharan Africa demonstrated a trend of remained stable or minor decrease. In contrast, regions such as Western Europe, Southern Latin America, and East Asia exhibited a significant decrease, indicating a marked reduction in the SAH burden (Fig 2, S3 Table). Notably, among the 204 countries and territories, Zimbabwe, Lesotho, Mozambique, and Turkmenistan exhibit a significant increase in SAH burden (S1 Fig, S4 Table).

## Decomposition analysis

The changes in the number of incident cases, mortality, and DALYs of SAH among AYAs from 1990 to 2021 can be attributed to three primary factors: population growth, population aging, and epidemiological change. The global increase in SAH incidence among AYAs (12.56%) was largely driven by population growth (32.77%) and aging (6.41%). However, epidemiological changes (-26.62%) led to a decline, partially offsetting the overall increase (Fig 3A). The growth in global prevalence exhibited a pattern similar to that of incidence (Fig 3B). The global decline in mortality (-26.63%) was influenced by population growth (27.74%) and aging (6.33%), with epidemiological changes being the primary factor for the decline (-60.7%)

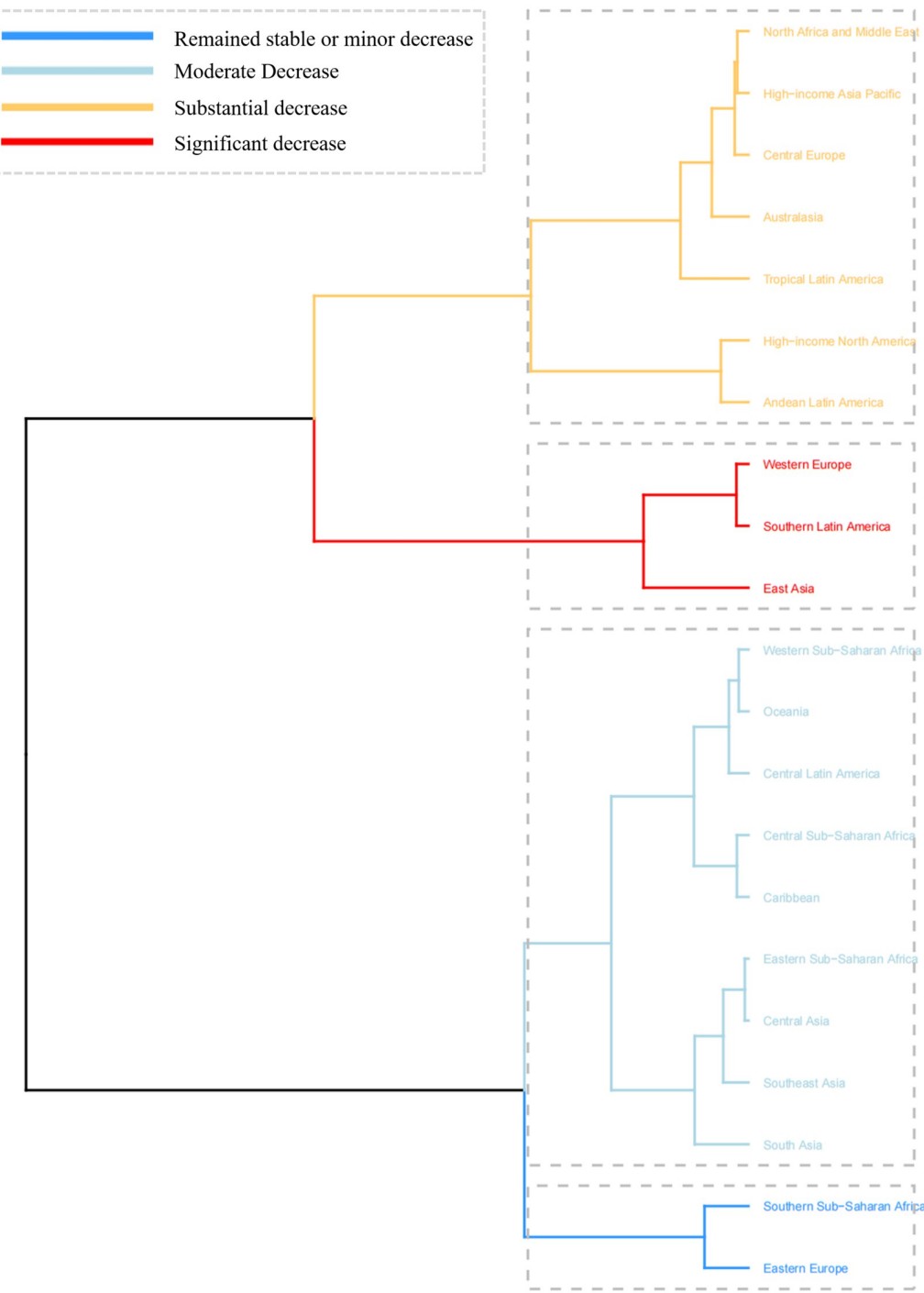

**Fig 2. Cluster analysis results based on the AAPC values of the age-standardized rates for deaths and DALYs attributed to SAH burden in the global AYA population from 1990 to 2021. Notes:** This figure presents the hierarchical clustering of regions based on the AAPC in age-standardized rates for deaths and DALYs attributed to SAH. Regions are classified into four categories: remained stable or minor decrease (blue), moderate decrease (light blue), substantial decrease (yellow), and significant decrease (red). **Abbreviations:** AAPC: estimated annual percentage change; DALYs: disability-adjusted-life-years.

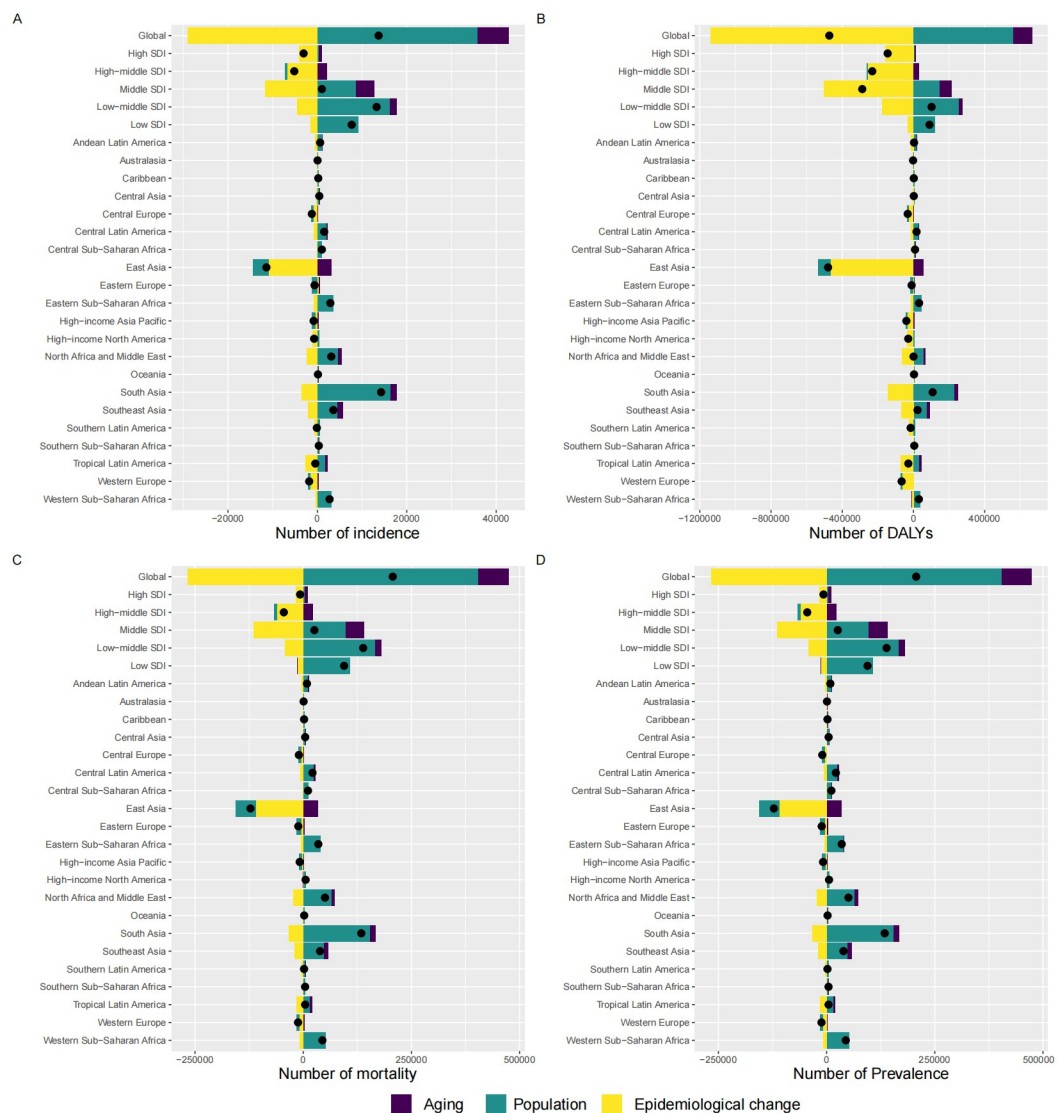

**Fig 3. Decomposition analysis of SAH burden in global AYAs population from 1990 to 2021. Notes:** This image presents a decomposition analysis of the burden of SAH among AYAs globally from 1990 to 2021. The analysis breaks down the changes in incidence, prevalence, mortality, and DALYs into three contributing factors: population growth, population aging, and epidemiological changes. **A:** Decomposition of SAH Incidence. **B:** Decomposition of SAH Prevalence. **C:** Decomposition of SAH Mortality. **D:** Decomposition of SAH DALYs. The black dot represents the net change resulting from all three components combined. For individual components, a positive value indicates an increase in SAH case numbers associated with that component, while a negative value denotes a corresponding decrease. The magnitude of each value reflects the extent of impact on SAH case numbers. **Abbreviations:** SAH: Subarachnoid Hemorrhage; DALYs: Disability-Adjusted Life Years; SDI: Sociodemographic Index.

(Fig 3C). Similarly, the reduction in global DALYs mirrored the pattern observed in deaths (Fig 3D).

Among the five SDI regions, High and High-middle SDI regions demonstrate an overall decrease in incidence, prevalence, deaths, and DALYs, primarily attributable to epidemiological improvements. Conversely, Low-middle and Low SDI areas exhibit increases in all metrics, mainly due to population growth. Middle SDI regions present a nuanced picture: slight

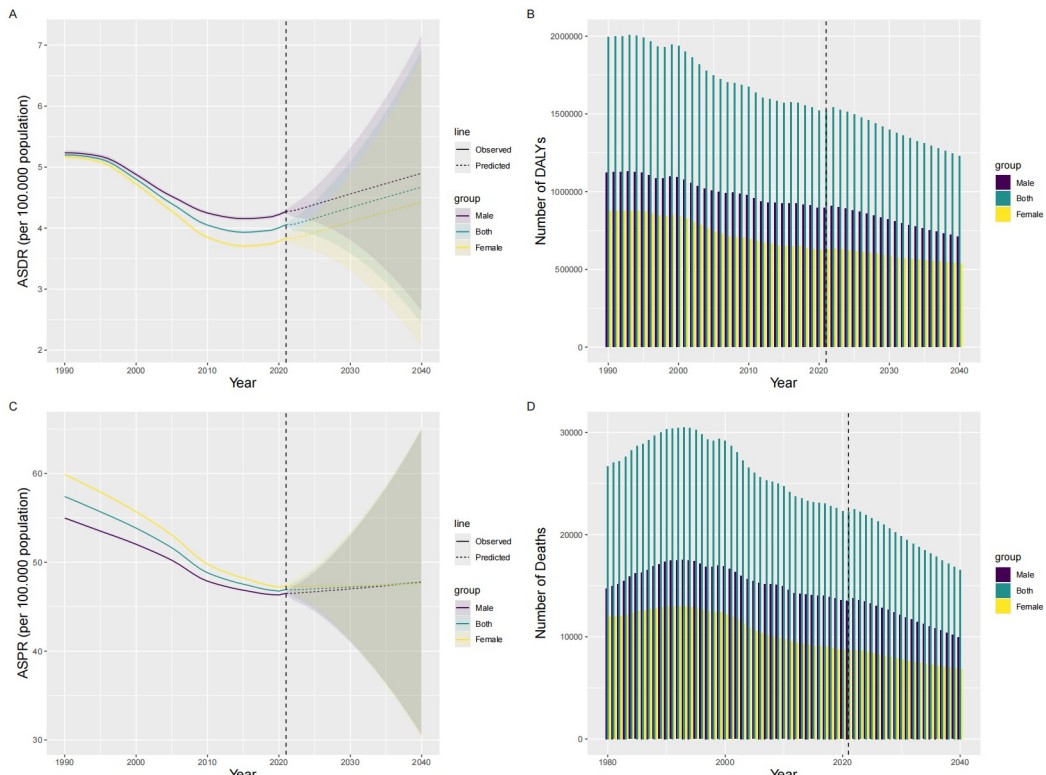

**Fig 4. Projections of SAH burden in global AYAs population from 1990 to 2040. Notes:** This image presents the observed and projected trends of SAH burden among AYAs globally from 1990 to 2040. The projections include ASR of incidence and prevalence, as well as the number of DALYs and deaths, segmented by gender (male, female, and both). **A:** ASR of SAH Incidence. **B:** Number of DALYs due to SAH. **C:** ASR of SAH Prevalence. **D:** Number of Deaths due to SAH. **Abbreviations:** SAH: Subarachnoid Hemorrhage; ASR: Age-Standardized Rate; DALYs: Disability-Adjusted Life Years.

increases in incidence and prevalence driven by population growth, yet decreases in deaths and DALYs owing to epidemiological advancements (S5 Table).

Epidemiological changes were the main factors for the decrease in SAH incidence among AYAs in regions such as Central Europe, East Asia, High-income North America, Southern Latin America, Tropical Latin America, and Western Europe. On the other hand, population growth was a significant factor for increased incidence in regions like Central Sub-Saharan Africa, Eastern Sub-Saharan Africa, Oceania, South Asia, and Western Sub-Saharan Africa (S5 Table).

At the regional level, Central Sub-Saharan Africa saw the most significant overall increases in incidence, prevalence, mortality, and DALYs for SAH among AYAs, with increases of 160.37%, 132.05%, 155.06%, and 133.56%, respectively.

## Prediction of global trends

From 2021 to 2040, it is projected that the ASIR and ASPR of SAH among AYAs will see a slight increase globally, while the ASMR and ASDR will decline. Gender differences will persist, with men experiencing a higher ASIR and greater disease burden compared to women (Fig 4).

Globally, the ASIR is expected to reach 4.67 per 100,000 people, resulting in approximately 158,727 new cases by 2040. The ASPR is projected to be 47.80 per 100,000 people, amounting

to about 1,599,393 prevalent cases by 2040. Notably, the global ASMR is anticipated to decrease to approximately 0.482 per 100,000 people, with the total number of deaths projected to reach 16,564 by 2040. Similarly, the ASDR is expected to drop to around 36 per 100,000 people, leading to an estimated 1,230,617 DALYs by 2040 (S6 Table).

## Discussion

In this study, we systematically demonstrated the spatial and temporal trends of subarachnoid hemorrhage among adolescents and young adults worldwide, covering incidence rates, prevalence rates, mortality numbers, and DALYs from 1990 to 2021. Our research found that since 1990, the burden of SAH among AYAs globally has significantly increased, with notable spatiotemporal variations across different SDI regions. Although the absolute number of SAH cases and prevalence has increased globally, the ASIR and ASPR have shown a decreasing trend. Similarly, the number of deaths and DALYs also exhibited a significant downward trend. This phenomenon is particularly evident in high and high-middle SDI regions, while low and low-middle-SDI regions displayed a higher disease burden. Our analysis reveals that Middle-SDI regions exhibited higher incidence and prevalence rates of SAH compared to Low-SDI regions, corroborating the findings of Lv B et al. regarding the greater SAH disease burden in Middle-SDI regions [15]. This observation can be attributed to a complex interplay of factors characteristic of these transitioning economies. Middle-SDI regions are undergoing a rapid epidemiological transition, marked by evolving healthcare systems and significant lifestyle changes. While these areas have developed better diagnostic capabilities than their low-SDI counterparts, they often lack the advanced treatment facilities and specialized care available in high-SDI regions [16]. This disparity in healthcare infrastructure may lead to increased detection of SAH cases without commensurate improvement in treatment outcomes, potentially contributing to higher prevalence rates. Concurrently, these regions are experiencing rapid urbanization, which has precipitated substantial lifestyle changes. The urban environment often fosters increased psychosocial stress, more sedentary behaviors, and the adoption of unhealthy dietary habits—all recognized risk factors for SAH [17]. The confluence of these urbanization-related factors may exacerbate the incidence of SAH in these transitioning economies. Gender disparities further complicate this landscape. While women generally demonstrate a higher propensity for seeking preventive healthcare services [18], potentially facilitating earlier detection and management of SAH risk factors, they may also face significant sociocultural barriers to accessing healthcare in certain contexts. In some societies, women may prioritize family care responsibilities over personal health needs [19]. These contrasting dynamics create a complex interplay that influences gender-specific SAH patterns in middle-SDI regions. The combination of transitioning healthcare systems, urbanization-related lifestyle changes, and gender-specific factors thus presents a unique challenge in addressing the SAH burden in these evolving economies. This complex scenario likely contributes to the observed higher incidence and prevalence rates in middle and low-middle SDI regions, highlighting the need for targeted interventions that address these multifaceted challenges.

Since 1990, the global burden of SAH among AYAs has significantly increased, though the ASIR and ASPR have both shown a declining trend. Specifically, the ASIR decreased from a high level in 1990 to 2014, followed by an inflection point, and then began to rise, accelerating in 2019. The analysis indicates that the overall decline in ASIR and ASPR can be attributed to multiple factors. Firstly, advancements in medical technology and public health interventions are important drivers [20, 21]. Early detection and intervention for potential aneurysms and other causes of SAH before they occur have become more feasible. Additionally, global public

health interventions have facilitated this process. For example, since 1998, the WHO has consistently emphasized preventive measures like reducing smoking, unhealthy diets, and physical inactivity through international consensus resolutions on the prevention and control of non-communicable diseases, including cardiovascular diseases [22–24].

Despite the overall decline in ASIR from 1990 to 2014, the ASIR experienced an inflection point and began to rise from 2014, with accelerated growth in 2019. Ezzati et al. pointed out that the continuous increase in global obesity and overweight rates is a significant factor [25]. With the proliferation of fast food culture and unhealthy dietary habits, obesity rates have risen significantly, thereby increasing the risk of SAH. Additionally, air pollution and other environmental pollutants have worsened in certain regions, which may also elevate the risk of cardiovascular diseases, including SAH [26]. In 2019, the outbreak of the COVID-19 pandemic exacerbated the pressure on global healthcare systems, with many countries reallocating medical resources to address the pandemic, leading to reduced resources for diagnosing and treating other diseases, such as SAH [27]. There have also been reports indicating the neurotropic properties of the COVID-19 virus showing correlations with the occurrence of cerebrovascular accidents, and it has been reported that vaccination against COVID-19 has exacerbated the management outcomes for stroke patients [28, 29]. For example, delays in both emergency and non-emergency surgeries could negatively impact the timely diagnosis and treatment of SAH patients, particularly among men who are less likely to seek medical help [30]. Furthermore, the economic pressure and social instability during the pandemic have increased psychological and physiological stress for many individuals, potentially further elevating the risk of SAH [31, 32].

When examining specific youth populations, George A. Mensah's research reveals that metabolic risk factors, particularly high systolic blood pressure, and dietary factors significantly contribute to SAH etiology, with a more pronounced impact on the elderly [33]. However, for younger demographics, lifestyle patterns emerge as more critical determinants. Beneficial dietary habits, regular physical activity, and adequate sleep are linked to lower systolic blood pressure in younger individuals [34]. Yet, factors such as smoking, unhealthy food choices, and stress increase their susceptibility to cardiovascular diseases [35]. This paradox adds complexity to understanding SAH risk in youth. In terms of diet, the intake of fruits and vegetables among adolescents is generally insufficient, while the consumption of carbonated soft drinks and fast food increases the risk of cardiovascular diseases [36]. These factors underscore the importance of focusing on the unique lifestyle patterns of younger populations and highlight the urgency of developing appropriate public health strategies to address these issues.

Our study found that the burden of SAH is higher in men, which is consistent with extensive published evidence indicating that the burden of cardiovascular diseases, including stroke, is generally lower among women [37]. This gender disparity has been well-documented in the literature, and our findings align with these established trends.

The ASIR and ASPR for males are higher than for females, which can be attributed to several factors. Firstly, males are more likely to engage in high-risk behaviors such as smoking and excessive alcohol consumption, both of which are significant risk factors for SAH [38]. Studies have shown that smoking and alcohol consumption rates are significantly higher among men than women, which may lead to a higher incidence of SAH among men [39]. Additionally, men often bear more economic and social pressures in many societies, which can lead to increased hypertension and other cardiovascular issues [40]. Secondly, Fuentes AM's study demonstrates a higher prevalence of aneurysms and SAH among older females, suggesting that postmenopausal estrogen decline may increase aneurysm risk. Our research on premenopausal women epidemiologically validates estrogen's protective effect against aneurysm formation, offering insights into hormonal influences on SAH risk across women's life

stages [41]. Estrogen's protective role in the cardiovascular system is attributed to its promotion of nitric oxide (NO) production, which maintains vascular elasticity and function while reducing inflammatory responses and oxidative stress [42]. These findings underscore the significance of sex hormones in SAH risk assessment and prevention strategies for women at different ages. Conversely, testosterone may increase the level of angiotensin II, promoting vasoconstriction and vascular stiffness, thereby increasing the risk of SAH [43]. Furthermore, men are generally less likely to seek medical help and have lower health awareness, which may lead to delayed medical consultations when early symptoms appear, thus increasing the incidence of SAH [44]. In contrast, women are more likely to seek medical help and undergo health check-ups, which may aid in the early detection and management of risk factors [45].

The decomposition analysis confirms that the increase in the number of SAH incidents and prevalent cases among the global AYA population from 1990 to 2021 is primarily driven by population growth. Conversely, the decline in the number of deaths and DALYs is mainly attributed to epidemiological changes. This highlights the effectiveness of public health strategies and preventive measures, such as surveillance and early detection initiatives. In the analysis of changes in incidence, prevalence, mortality, and DALYs across different SDI countries, epidemiological changes have made substantial contributions in middle-SDI regions. This underscores the significant impact of epidemiological shifts in countries with moderate socio-demographic indices, suggesting that well-formulated health strategies can have more pronounced effects in these regions. On the other hand, as the Socio-Demographic Index decreases, population growth becomes a more prominent factor contributing to disease burden. In low-SDI countries, population growth remains the primary contributor to the disease burden. These findings suggest that when formulating cerebrovascular public health strategies, it is crucial to focus on low-SDI countries [46]. Appropriate public health strategies have greater potential for impact in these regions.

Our predictions are based on GBD data collected before and during the COVID-19 pandemic (2019–2021) [47], introducing potential biases due to the pandemic's impact on non-communicable diseases and healthcare systems [48]. The pandemic's economic consequences may have increased psychological stress among young people, potentially raising SAH incidence rates [49]. Additionally, isolation policies often prevented timely access to critical medical interventions [50]. Given these considerations, we limited our forecast to 2040, acknowledging the uncertainty introduced by recent pandemic-related data. As our understanding of COVID-19 and its aftermath evolves, future GBD updates may enable more accurate long-term predictions. Our projections suggest that without intervention, SAH incidence among AYAs will increase over the next 20 years. This underscores the urgent need for further research, focusing on both early clinical screening and tertiary prevention. As SAH continues to pose a significant neurosurgical burden, a multifaceted approach to prevention, early detection, and treatment is essential to mitigate its impact on AYAs in the coming decades.

Our study is uniquely positioned within the critical timeframe encompassing the COVID-19 pandemic, extending the analysis beyond 2019. This temporal context is crucial, as it allows us to examine SAH trends against the backdrop of unprecedented global health challenges. Furthermore, our research specifically targets adolescent populations, an age group often underrepresented in SAH studies. By focusing on this demographic during such a pivotal period, we address a significant gap in the existing literature.

In conclusion, our study's distinct combination of adolescent focus and post-2019 data provides essential insights for developing targeted SAH prevention strategies and epidemiological policies. These findings are particularly valuable in the context of long COVID, offering crucial guidance for mitigating SAH risks in young populations as we navigate the evolving post-pandemic landscape.

## Supporting information

**S1 Table. Joinpoint analysis of age-standardized rates of AYA subarachnoid Hemorrhage from 1990 to 2021.**
(XLS)

**S2 Table. Subarachnoid hemorrhage trends from 1990 to 2021 by region.**
(XLS)

**S3 Table. AAPC for both sexes in 21 regions from 1990 to 2021.**
(XLS)

**S4 Table. AAPC for both sexes in 204 countries and territories from 1990 to 2021.**
(XLS)

**S5 Table. Decomposition analysis data for the incidence, prevalence, deaths, and DALYs of SAH in the AYA population between 1990 and 2021.**
(XLS)

**S6 Table. BAPC prediction of SAH cases and rates among AYAs from 2021 to 2040.**
(XLS)

**S1 Fig. Cluster analysis results based on the AAPC values of the age-standardized rates for deaths and DALYs attributed to SAH burden in 204 countries and territories from 1990 to 2021.**
(TIF)

## Acknowledgments

We extend our gratitude to the collaborators of the Global Burden of Diseases, Injuries, and Risk Factors Study 2021 for their exceptional work. We appreciate Jieni Wu for her valuable suggestions on the data formatting in this paper. And we are deeply grateful to Dr. Yifei Su for his valuable suggestions that significantly improved this manuscript.

## Author Contributions

**Conceptualization:** Xuanchen Liu, Yingda Song, Xiaoxiong Yang, Xiaochen Niu.

**Data curation:** Xuanchen Liu, Yingda Song.

**Formal analysis:** Xuanchen Liu, Rui Cheng.

**Funding acquisition:** Chunhong Wang, Guijun Jia, Hongming Ji.

**Investigation:** Xuanchen Liu.

**Methodology:** Xuanchen Liu, Rui Cheng, Xiaochen Niu.

**Software:** Xiaochen Niu.

**Supervision:** Chunhong Wang, Guijun Jia, Hongming Ji.

**Validation:** Xuanchen Liu, Hongming Ji.

**Visualization:** Xuanchen Liu.

**Writing – original draft:** Xuanchen Liu, Yingda Song.

**Writing – review & editing:** Xuanchen Liu, Xiaoxiong Yang.

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
