## [Decision Letter · Decision Letter 0]

15 Aug 2024

PONE-D-24-29086Global burden of subarachnoid hemorrhage among adolescents and young adults aged 15–39 years: A trend analysis study from 1990 to 2021PLOS ONE

Dear Dr. Ji,

Thank you for submitting your manuscript to PLOS ONE. After careful consideration, we feel that it has merit but does not fully meet PLOS ONE’s publication criteria as it currently stands. Therefore, we invite you to submit a revised version of the manuscript that addresses the points raised during the review process.

We look forward to receiving your revised manuscript.

Kind regards,

Muhammad Fawad, PhD

Academic Editor

PLOS ONE

Journal Requirements:

This work was financially supported by the Basic Research Plan of Shanxi Province in 2021, grant number 202103021224386, awarded to Dr. Hongming Ji. The funding body played no role in the study design, data collection, data analysis, decision to publish, or preparation of the manuscript. For more information about the funding source, please visit the official website at https://kjt.shanxi.gov.cn/.

5. Ethics statement appears in the Methods section of the manuscript AND at the end of the manuscript:

Your ethics statement should only appear in the Methods section of your manuscript. If your ethics statement is written in any section besides the Methods, please delete it from any other section. 

6. We note that Figure S1 in your submission contain map images which may be copyrighted. All PLOS content is published under the Creative Commons Attribution License (CC BY 4.0), which means that the manuscript, images, and Supporting Information files will be freely available online, and any third party is permitted to access, download, copy, distribute, and use these materials in any way, even commercially, with proper attribution. For these reasons, we cannot publish previously copyrighted maps or satellite images created using proprietary data, such as Google software (Google Maps, Street View, and Earth). For more information, see our copyright guidelines: http://journals.plos.org/plosone/s/licenses-and-copyright.

We require you to either present written permission from the copyright holder to publish these figures specifically under the CC BY 4.0 license, or remove the figures from your submission:

a. You may seek permission from the original copyright holder of Figure S1 to publish the content specifically under the CC BY 4.0 license.  

**Additional Editor Comments:**

Authors should provide more insights into the specific socio-economic or healthcare factors that might help clarify why Middle-SDI regions experience a higher burden than Low-SDI regions?

Did the authors consider the impact of gender-specific healthcare access and health-seeking behaviors when explaining the observed disparities between men and women?

The figures in the manuscript are blurry and difficult to interpret. Please improve the resolution of these figures or consider redrawing them to ensure a clear presentation and easy understanding for readers.

Reviewers' comments:

Reviewer's Responses to Questions

**Comments to the Author**

1. Is the manuscript technically sound, and do the data support the conclusions?

Reviewer #1: Yes

Reviewer #2: Yes

Reviewer #3: Yes

2. Has the statistical analysis been performed appropriately and rigorously? 

Reviewer #1: Yes

Reviewer #2: Yes

Reviewer #3: Yes

3. Have the authors made all data underlying the findings in their manuscript fully available?

Reviewer #1: Yes

Reviewer #2: Yes

Reviewer #3: Yes

4. Is the manuscript presented in an intelligible fashion and written in standard English?

Reviewer #1: Yes

Reviewer #2: Yes

Reviewer #3: Yes

5. Review Comments to the Author

**Reviewer #1: **Strengths: The study has a clear and specific objective: analyzing the global burden of subarachnoid hemorrhage (SAH) among adolescents and young adults (AYAs) aged 15–39 years over a 31-year period, focusing on spatial and temporal trends. This provides a targeted approach, which is beneficial for creating specialized public health strategies.

Opportunities for Improvement: The objective could be enhanced by mentioning the geographical scope and the importance of the study in the context of global health explicitly. Additionally, the rationale for selecting the age group 15-39 years should be briefly explained to highlight its significance.

Methods

Strengths:

The study utilizes data from the Global Burden of Disease Study 2021, which is a comprehensive and widely recognized source.

Segmenting the data by age groups and socio-demographic index (SDI) quintiles allows for a detailed analysis of different populations.

The use of statistical methods such as Joinpoint regression and decomposition analysis adds robustness to the analysis of temporal trends and epidemiological changes.

Opportunities for Improvement:

The methods section could benefit from a more detailed explanation of the statistical techniques used, particularly Joinpoint regression and decomposition analysis, to aid readers who may not be familiar with these methods.

Including a brief discussion on data limitations, such as potential biases or gaps in the GBD 2021 data, would provide a more balanced view of the study's methodology.

Results

Strengths:

The results are presented with clear numerical data, showing both absolute numbers and age-standardized rates, which enhances the comprehensiveness of the analysis.

Highlighting the decrease in mortality and DALYs alongside the increase in incidence and prevalence provides a nuanced understanding of the burden of SAH among AYAs.

The regional analysis adds valuable insights into where the burden is most substantial, which is crucial for targeted interventions.

Opportunities for Improvement:

The results section could be improved by including visual aids such as graphs or charts to illustrate trends more effectively.

Providing a more in-depth analysis of why certain trends occurred, especially the increase in incidence and prevalence in recent years, would add depth to the findings.

Discussion and Conclusion

Strengths:

The discussion appropriately emphasizes the need for targeted public health interventions in low and low-middle SDI regions.

The conclusion ties the findings to practical recommendations, such as improving healthcare resources and preventive strategies, making the study's implications clear and actionable.

Opportunities for Improvement:

The discussion could benefit from a comparison with other studies or existing literature to place the findings within a broader context.

Expanding on potential future research directions or unanswered questions that arose from the study would provide a pathway for further investigation and enhance the overall impact of the research.

Overall Assessment

The study provides a comprehensive analysis of the global burden of subarachnoid hemorrhage among adolescents and young adults, with clear strengths in data utilization, statistical analysis, and practical recommendations. However, there are opportunities to improve the clarity and depth of the methodology, results, and discussion sections. Enhancing these aspects would strengthen the study's contribution to public health knowledge and policy formulation.

**Reviewer #2: **I appreciate the opportunity to critically review the manuscript PONE-D-24-29086, in which the authors analyze the burden of SAH among adolescents and young adults using the most recent GBD data (2021). The manuscript is clear, and the methodology is robust. I present a series of minor comments for the authors' consideration.

Comment 1: Lines 370-371: The authors indicate that they found the burden of SAH to be higher in men. While this is true, the current wording makes it appear that this is a unique finding of the study, which is not correct. There is extensive published evidence indicating that the burden of cardiovascular diseases is generally lower among women. I suggest briefly mentioning this.

Comment 2: Published evidence (DOI: 10.1016/j.jacc.2023.11.007) suggests that metabolic risk factors (i.e., high systolic blood pressure) and dietary factors have a significant attributable fraction in SAH, and in general (as expected) this was higher in older age groups. Although the authors did not aim to analyze risk factors, it would be interesting if they briefly discussed the impact these risk factors might have on this particular age group, composed of young individuals.

Comment 3: Figures 1-3 were illegible. I would appreciate it if they could be presented clearly should I have the pleasure of being considered as a reviewer for the revised version of the document.

Comment 4: The GBD 2021 Forecasting Collaborators Group conducted analyses up to the year 2050. Is there any particular methodological reason for conducting the analysis up to the year 2040? I suggest specifying this in the Discussion section.

**Reviewer #3:** This is a manuscript with interesting title that written well.

There are only some minor mistakes in some lines (including 197, 262, …) that can be corrected.

Also the quality of figures are very low that need to improve.

I suggest that in the decomposition analysis, the clustering analysis was performed and the characteristics of clusters across on the SDI categories, region, … was assessed.

6. PLOS authors have the option to publish the peer review history of their article (what does this mean?). If published, this will include your full peer review and any attached files.

Reviewer #1: No

Reviewer #2: **Yes: **Efrén Murillo-Zamora

Reviewer #3: **Yes: **Farzane Ahmadi

---

## [Author Response · Author response to Decision Letter 0]

4 Sep 2024

4 September 2024

Muhammad Fawad, PhD

Academic Editor

PLOS ONE

Dear Editor:

We really appreciate the critical reading of our manuscript entitled “Global burden of subarachnoid hemorrhage among adolescents and young adults aged 15–39 years: A trend analysis study from 1990 to 2021” (Manuscript Number: PONE-D-24-29086R1). We thank you for all the valuable suggestions received from the reviewers and editor. We have carefully considered the comments and have revised the manuscript accordingly. The responses to the comments are listed one by one as follows (please see below). We are looking forward to hearing about your decision.

Journal Requirements:

Answer: Thank you for your comments. I have made the necessary formatting revisions as per your requirements and have resubmitted the manuscript.

This work was financially supported by the Basic Research Plan of Shanxi Province in 2021, grant number 202103021224386, awarded to Dr. Hongming Ji. The funding body played no role in the study design, data collection, data analysis, decision to publish, or preparation of the manuscript. For more information about the funding source, please visit the official website at https://kjt.shanxi.gov.cn/.

Answer: Thank you for your correction regarding the accuracy of my Funding Statement. I have now uploaded a revised version that includes additional details about other sources of financial support received during this study.

This work was funded by the Basic Research Plan of Shanxi Province in 2021 (202103021224386) and the 2024 Graduate Education Innovation Plan of Shanxi Province (2024SJ177). The funders had no role in study design, data collection and analysis, decision to publish, or preparation of the manuscript. There was no additional external funding received for this study.

3.When completing the data availability statement of the submission form, you indicated that you will make your data available on acceptance. We strongly recommend all authors decide on a data sharing plan before acceptance, as the process can be lengthy and hold up publication timelines. Please note that, though access restrictions are acceptable now, your entire data will need to be made freely accessible if your manuscript is accepted for publication. This policy applies to all data except where public deposition would breach compliance with the protocol approved by your research ethics board. If you are unable to adhere to our open data policy, please kindly revise your statement to explain your reasoning and we will seek the editor's input on an exemption. Please be assured that, once you have provided your new statement, the assessment of your exemption will not hold up the peer review process.

Answer: All relevant data are within the manuscript and its Supporting Information files.

4.PLOS requires an ORCID iD for the corresponding author in Editorial Manager on papers submitted after December 6th, 2016. Please ensure that you have an ORCID iD and that it is validated in Editorial Manager. To do this, go to ‘Update my Information’ (in the upper left-hand corner of the main menu), and click on the Fetch/Validate link next to the ORCID field. This will take you to the ORCID site and allow you to create a new iD or authenticate a pre-existing iD in Editorial Manager. Please see the following video for instructions on linking an ORCID iD to your Editorial Manager account: https://www.youtube.com/watch?v=_xcclfuvtxQ

Answer: Thank you for your guidance. I have now uploaded the ORCID iD for the corresponding author and completed the necessary information update.

5.Ethics statement appears in the Methods section of the manuscript AND at the end of the manuscript:

Your ethics statement should only appear in the Methods section of your manuscript. If your ethics statement is written in any section besides the Methods, please delete it from any other section. 

Answer: Thank you for your feedback regarding the placement of the ethics statement. I have now moved the ethics statement to the Methods section of the manuscript.

6.We note that Figure S1 in your submission contain map images which may be copyrighted. All PLOS content is published under the Creative Commons Attribution License (CC BY 4.0), which means that the manuscript, images, and Supporting Information files will be freely available online, and any third party is permitted to access, download, copy, distribute, and use these materials in any way, even commercially, with proper attribution. For these reasons, we cannot publish previously copyrighted maps or satellite images created using proprietary data, such as Google software (Google Maps, Street View, and Earth). For more information, see our copyright guidelines: http://journals.plos.org/plosone/s/licenses-and-copyright.

We require you to either present written permission from the copyright holder to publish these figures specifically under the CC BY 4.0 license, or remove the figures from your submission:

a. You may seek permission from the original copyright holder of Figure S1 to publish the content specifically under the CC BY 4.0 license.  

Answer: Thank you for your comments and for emphasizing the importance of map copyright. Considering the copyright concerns and the variations in national boundaries across different countries and regions, as well as the fact that the original Figure S1 was supplementary to Table 1 in the manuscript, we have decided to remove Figure S1. This removal does not alter the main discussion of the manuscript, and the remaining tables can adequately provide the necessary detailed data.

7.Please review your reference list to ensure that it is complete and correct. If you have cited papers that have been retracted, please include the rationale for doing so in the manuscript text, or remove these references and replace them with relevant current references. Any changes to the reference list should be mentioned in the rebuttal letter that accompanies your revised manuscript. If you need to cite a retracted article, indicate the article’s retracted status in the References list and also include a citation and full reference for the retraction notice.

Answer: We have carefully reviewed and verified that our reference list does not include any retracted papers. The references we removed have been marked in red, and the new references we added have been highlighted in blue.

Additional Editor Comments:

Authors should provide more insights into the specific socio-economic or healthcare factors that might help clarify why Middle-SDI regions experience a higher burden than Low-SDI regions?

Answer: Thank you for your suggestion regarding the discussion. I have added a discussion on the specific characteristics and higher burden in Middle-SDI regions compared to Low-SDI and High-SDI countries, with a focus on their unique socio-economic factors and healthcare challenges.

Manuscripts Page 21-22 Line 354--369

“Our analysis reveals that Middle-SDI regions exhibited higher incidence and prevalence rates of SAH compared to Low-SDI regions, corroborating the findings of Lv B et al. regarding the greater SAH disease burden in Middle-SDI regions[15]. This observation can be attributed to a complex interplay of factors characteristic of these transitioning economies. Middle-SDI regions are undergoing a rapid epidemiological transition, marked by evolving healthcare systems and significant lifestyle changes. While these areas have developed better diagnostic capabilities than their low-SDI counterparts, they often lack the advanced treatment facilities and specialized care available in high-SDI regions [16]. This disparity in healthcare infrastructure may lead to increased detection of SAH cases without commensurate improvement in treatment outcomes, potentially contributing to higher prevalence rates. Concurrently, these regions are experiencing rapid urbanization, which has precipitated substantial lifestyle changes. The urban environment often fosters increased psychosocial stress, more sedentary behaviors, and the adoption of unhealthy dietary habits - all recognized risk factors for SAH [17]. ”

Did the authors consider the impact of gender-specific healthcare access and health-seeking behaviors when explaining the observed disparities between men and women?

Answer: Thank you for your valuable feedback. The impact of gender-specific healthcare access and health-seeking behaviors was indeed an aspect that was previously lacking in our discussion. I have now expanded the discussion in the revised manuscript to include a more detailed analysis of the differences in healthcare access and health-seeking behaviors between men and women.

Manuscripts Page 22 Line 369--383

“The confluence of these urbanization-related factors may exacerbate the incidence of SAH in these transitioning economies. Gender disparities further complicate this landscape. While women generally demonstrate a higher propensity for seeking preventive healthcare services [18], potentially facilitating earlier detection and management of SAH risk factors, they may also face significant sociocultural barriers to accessing healthcare in certain contexts. In some societies, women may prioritize family care responsibilities over personal health needs [19]. These contrasting dynamics create a complex interplay that influences gender-specific SAH patterns in middle-SDI regions. The combination of transitioning healthcare systems, urbanization-related lifestyle changes, and gender-specific factors thus presents a unique challenge in addressing the SAH burden in these evolving economies. This complex scenario likely contributes to the observed higher incidence and prevalence rates in middle and low-middle SDI regions, highlighting the need for targeted interventions that address these multifaceted challenges.

The figures in the manuscript are blurry and difficult to interpret. Please improve the resolution of these figures or consider redrawing them to ensure a clear presentation and easy understanding for readers.

Answer: Thank you for your feedback. I have redrawn all the figures in the manuscript to meet the journal's requirements and ensure they are clearly visible for readers. However, please note that the PDF version generated by the Editorial Manager system may render the images less clear. To view or download high-resolution versions of the figures, you can click on the "Click here to access/download" link in the top right corner of each image. We apologize for any inconvenience this may cause.

Comments to the Author

1. Is the manuscript technically sound, and do the data support the conclusions?

Reviewer #1: Yes

Reviewer #2: Yes

Reviewer #3: Yes

2. Has the statistical analysis been performed appropriately and rigorously?

Reviewer #1: Yes

Reviewer #2: Yes

Reviewer #3: Yes

3.Have the authors made all data underlying the findings in their manuscript fully available?

Reviewer #1: Yes

Reviewer #2: Yes

Reviewer #3: Yes

4. Is the manuscript presented in an intelligible fashion and written in standard English?

Reviewer #1: Yes

Reviewer #2: Yes

Reviewer #3: Yes

Answer: We sincerely appreciate the reviewers' and editor's recognition of our data handling and the content of the manuscript. On behalf of all the authors, I would like to thank you for your thorough review and constructive feedback. We will carefully revise the manuscript according to your professional and insightful comments. Once again, thank you for your expert review and valuable input.

5. Review Comments to the Author

Reviewer #1: Strengths: The study has a clear and specific objective: analyzing the global burden of subarachnoid hemorrhage (SAH) among adolescents and young adults (AYAs) aged 15–39 years ov

---

## [Decision Letter · Decision Letter 1]

14 Oct 2024

PONE-D-24-29086R1Global burden of subarachnoid hemorrhage among adolescents and young adults aged 15–39 years: A trend analysis study from 1990 to 2021PLOS ONE

Dear Dr. Ji,

Thank you for submitting your manuscript to PLOS ONE. After careful consideration, we feel that it has merit but does not fully meet PLOS ONE’s publication criteria as it currently stands. Therefore, we invite you to submit a revised version of the manuscript that addresses the points raised during the review process.

We look forward to receiving your revised manuscript.

Kind regards,

Muhammad Fawad, PhD

Academic Editor

PLOS ONE

Journal Requirements:

Additional Editor Comments:

The authors' response effectively addresses both the reviewer's and my comments; however, some concerns from the reviewer still remain, and these should be clarified.

Reviewers' comments:

Reviewer's Responses to Questions

**Comments to the Author**

1. If the authors have adequately addressed your comments raised in a previous round of review and you feel that this manuscript is now acceptable for publication, you may indicate that here to bypass the “Comments to the Author” section, enter your conflict of interest statement in the “Confidential to Editor” section, and submit your "Accept" recommendation.

Reviewer #1: All comments have been addressed

Reviewer #2: All comments have been addressed

Reviewer #3: (No Response)

2. Is the manuscript technically sound, and do the data support the conclusions?

Reviewer #1: Yes

Reviewer #2: Yes

Reviewer #3: Yes

3. Has the statistical analysis been performed appropriately and rigorously? 

Reviewer #1: Yes

Reviewer #2: Yes

Reviewer #3: Yes

4. Have the authors made all data underlying the findings in their manuscript fully available?

Reviewer #1: Yes

Reviewer #2: Yes

Reviewer #3: Yes

5. Is the manuscript presented in an intelligible fashion and written in standard English?

Reviewer #1: Yes

Reviewer #2: Yes

Reviewer #3: Yes

6. Review Comments to the Author

Reviewer #1: The article presents interesting results of great importance for public health. The authors have made the necessary changes. The article can be accepted for publication.

Reviewer #2: I thank the group of authors for considering the observations and suggestions I previously provided. I congratulate them on their interesting manuscript. It was a pleasure to critically review it.

Reviewer #3: Unfortunately, there is no correction for the clustering analysis in the manuscript.

The aim of the clustering analysis is to detect homogenous subgroups of countries without considering the SDI categories.

7. PLOS authors have the option to publish the peer review history of their article (what does this mean?). If published, this will include your full peer review and any attached files.

Reviewer #1: **Yes: **Jonas Wolf

Reviewer #2: **Yes: **Efrén Murillo-Zamora

Reviewer #3: **Yes: **Farzane Ahmadi

---

## [Author Response · Author response to Decision Letter 1]

1 Nov 2024

Answer to reviewer #3: Thank you very much for your constructive suggestion. We apologize for not fully understanding the application of the clustering method initially. After your emphasis, we extensively reviewed relevant literature to gain a better understanding of this method. Consequently, we have incorporated a Hierarchical Cluster Analysis to assess the AAPC of deaths and DALYs across the 21 regions and 204 countries and territories mentioned in our study. This analysis aims to identify different patterns of disease burden changes. We have included the clustering results for the 21 regions in Figure 2 of the main text, and due to space constraints, the clustering of the 204 countries and territories is provided in Supplementary Figure S1. Additionally, we attempted to explain the different clustering patterns and have included detailed AAPC data in the supplementary materials. The Statistical Analysis section now describes our methodology. Since the primary focus of our paper is the overall global trends of adolescent SAH, we have utilized Hierarchical Cluster Analysis to broadly categorize similar patterns of change. We appreciate your suggestion and recognize that using cluster analysis in GBD database studies can open new research avenues. We will pay more attention to this clustering method to detect homogenous subgroups of countries without considering SDI categories in future research. Thank you once again for your invaluable input.

---

## [Editor Report · Decision Letter 2]

6 Dec 2024

Global burden of subarachnoid hemorrhage among adolescents and young adults aged 15–39 years: A trend analysis study from 1990 to 2021

PONE-D-24-29086R2

Dear Dr. Ji,

We’re pleased to inform you that your manuscript has been judged scientifically suitable for publication and will be formally accepted for publication once it meets all outstanding technical requirements.

Kind regards,

Muhammad Fawad, PhD

Academic Editor

PLOS ONE

Additional Editor Comments (optional):

The authors have thoroughly addressed the comments and provided clear, well-supported responses to each concern.
---

## [Editor Report · Acceptance letter]

11 Dec 2024

PONE-D-24-29086R2 

PLOS ONE

Dear Dr. Ji, 

I'm pleased to inform you that your manuscript has been deemed suitable for publication in PLOS ONE. Congratulations! Your manuscript is now being handed over to our production team.

Kind regards, 

on behalf of

Dr. Muhammad Fawad 

Academic Editor

PLOS ONE